# A Super-Aligned Driving Generalist In Your Cockpit

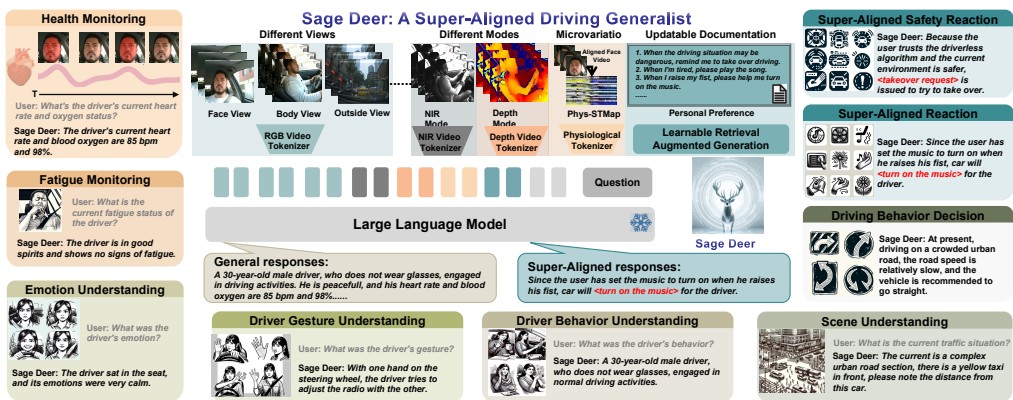

Figure 1: Sage Deer uses the pre-trained video encoders (visual tokenizers) to tokenize the different modes and views of the video, especially the physiological encoder used to extract the physiological signals of face video. Using tokenized visual embedding, large language models can provide general responses for physiological indicators, emotional states, gestures, human behavior, and scene understanding. It is worth emphasizing that the individual needs of the user can be edited in a single document. We then query this information with learnable retrieval augmentation generation, generating results super aligned with user preferences.

## ABSTRACT

The intelligent driving cockpit, an important part of intelligent driving, needs to match different users' comfort, interaction, and safety needs. This paper aims to build a **s**uper-**a**ligned and **ge**neralist **dr**iving agent, **sage deer**. Sage Deer achieves two highlights: (1) Super alignment: It achieves different reactions according to different people's preferences and biases. (2) Generalist: It can understand the user's physiological indicators, facial emotions, hand movements, body movements, driving scenarios, and behavioral decisions. (3) Multimodal: He can understand RGB, NIR, and depth video to build more robust perception, understanding, and reasoning. To achieve the above requirements, we design retrieval-enhanced multimodal frameworks. We collected multiple data sets and built a large-scale benchmark. This benchmark measures the deer's perceptual decision-making ability and the super alignment's accuracy.

## 1 INTRODUCTION

Owing to advancements in artificial intelligence and high performance computing hardware, intelligent connected vehicles have garnered significant attention from both academia and governmental bodies due to their profound potential to revolutionize future transportation paradigms Li et al. (2023b); Teng et al. (2023). By integrating advanced sensors, artificial intelligence algorithms, and technologies, s possess the capability to perceive their environment, make informed decisions, and execute control, which will significantly enhance vehicle safety, optimize traffic flow, and deliver a comfortable and convenient driving experience.

The intelligent driving cockpit serves as the interface for human interaction with s, offering a seamless and user-focused experience through the provision of real-time data, personalized settings, and

intuitive controls Li et al. (2023b); Yang et al. (2022). This advanced system enhances communication between the driver, passengers, and the vehicle, increases safety via comprehensive monitoring systems, and improves comfort by adapting to individual preferences and varying driving conditions.

However, the pursuit of a universal intelligent driving cockpit that encapsulates both general principles and individual-specific requirements is inherently complex. Individuals possess distinct driving habits and needs, necessitating the development of an agent that aligns with their preferences. However, it is impractical to train a unique model for each individual user. Existing research Cui et al. (2023b;a); Li et al. (2023c; 2022); Liu & Zhang (2020); Tao et al. (2024) on intelligent driving cockpits predominantly emphasizes rigid decision-making and control by integrating external environmental perception data and in-vehicle occupant status, which usually overlooks the individual preferences and limits the adaptability of cockpit system.

Recent advancements in multi-modal large models have demonstrated their capabilities in understanding and reasoning with multi-modal inputs, including videos and natural language. Sima et al. Sima et al. (2023) proposed the utilization of large language models to perform end-to-end autonomous driving through a visual question answering manner. While Wang et al. Wang et al. (2023) proposed DriveMLM, which can perform close-loop autonomous driving in realistic simulators by mitigate the gap between language decisions and vehicle control commands. As for in-vehicle situations, multi-modal large models are also applied to driver health monitoring Hecht et al. (2018), driving decision individuation with simple strategy Cui et al. (2023b). A robust and personalized intelligent driving cockpit can incorporate driving scenario understanding, language interaction, user behavior and sentiment analysis, processing of personalized driver needs, and intelligent decision-making. However, the above mentioned research has yet to achieve comprehensive functionality and full-modal information processing and reasoning. Thus, there remains a significant gap between current capabilities and the envisioned intelligent driving cockpit.

To bridge the gap between current approaches in providing a personalized driving experience and comprehensive omni-modal information processing, we propose a **super-a**ligned and **ge**neralist **dr**iving agent, **sage deer**. We present a retrieval-augmented framework designed to develop a super-aligned and generalist agent for smart cockpits. This framework is characterized by three core strengths: comprehensive tokenization of multi-mode and multi-view videos, robust generalist understanding, and super-aligned responses. To effectively integrate diverse visual and physiological data, we tokenize inputs from multiple sensing modalities—including RGB, near-infrared (NIR), and depth cameras—using a pre-trained ResNet18 for feature extraction, followed by mapping to a language space with a two-layer linear layer. Multi-view perspectives are similarly tokenized with distinct perspective markers to ensure comprehensive scene interpretation. Additionally, physiological information is processed through spatio-temporal representations and encoded into tokens, enabling the model to monitor driver states accurately. The framework employs a retrieval-augmented generation mechanism that leverages an updatable document containing both fundamental and personalized information, enhancing the large language model's (LLM) ability to generate contextually relevant responses without extensive fine-tuning. Expert knowledge fusion further integrates domain-specific insights, allowing the system to dynamically adapt based on real-time physiological and environmental data. This holistic approach ensures high-quality scene understanding, precise physiological monitoring, and personalized, context-aware interactions, significantly improving the alignment and generalization capabilities of agents within complex cockpit environments.

**Contribution**

1. Established a multi-view and multi-modal evaluation protocol for the intelligent cockpit, involving unified evaluation of the driver's physiology, emotion, behavior, driving scene understanding, and decision-making.

2. An intelligent driving cockpit super alignment evaluation protocol involving generalization ability for different needs was established.

3. Designed a set of algorithms that align human preferences without fine-tuning.

Table 1: Comparison of public driving cockpit perception datasets. "Naturalistic" refers to people's natural driving behavior, while "Induced" refers to people's intervention and design of scenes. Reaction is the response given by the intelligent cockpit according to different user needs. Question Answering refers to the verbal inquiry interaction between the user and the cockpit.

| Dataset | Views | Naturalistic | Induced | Multimodal | Physiological Indicator | Behavior | Emotion | Traffic Context | Vehicle Condition | Reaction | Question Answering |
|---|---|---|---|---|---|---|---|---|---|---|---|
| SEU Zhao et al. (2012) | 1 | – | ✓ | – | – | ✓ | – | – | – | – | – |
| Tran Tran et al. (2018) | 1 | – | ✓ | – | – | ✓ | – | – | – | – | – |
| Zhang Zhang et al. (2020) | 2 | – | ✓ | ✓ | – | ✓ | – | – | – | – | – |
| StateFarm sta (2016) | 1 | – | ✓ | – | – | ✓ | – | – | – | – | – |
| AUC-DD Eraqi et al. (2019) | 1 | ✓ | – | – | – | ✓ | – | – | – | – | – |
| LoLi Saad et al. (2020) | 1 | ✓ | – | ✓ | – | ✓ | – | – | – | – | – |
| Brain4Cars Jain et al. (2016) | 2 | ✓ | – | – | – | ✓ | – | – | – | – | – |
| Drive&Act Martin et al. (2019) | 6 | – | ✓ | ✓ | – | ✓ | – | – | – | – | – |
| DMD Ortega et al. (2020) | 3 | – | ✓ | ✓ | – | ✓ | – | – | – | – | – |
| DAD Kopuklu et al. (2021) | 2 | – | ✓ | ✓ | – | ✓ | – | – | – | – | – |
| DriPE Guesdon et al. (2021) | 1 | ✓ | – | – | – | – | – | – | – | – | – |
| LBW Kasahara et al. (2022) | 2 | ✓ | – | – | – | – | – | – | – | – | – |
| MDAD Jegham et al. (2019) | 2 | ✓ | – | ✓ | – | ✓ | – | – | – | – | – |
| 3MDAD Jegham et al. (2020) | 2 | ✓ | – | ✓ | – | ✓ | – | – | – | – | – |
| DEFE Li et al. (2021a) | 1 | – | ✓ | – | – | – | ✓ | – | – | – | – |
| DEFE+ Li et al. (2021b) | 1 | – | ✓ | ✓ | – | – | ✓ | – | – | – | – |
| Du Du et al. (2020) | 1 | – | ✓ | ✓ | – | – | ✓ | – | – | – | – |
| KMU-FED Jeong & Ko (2018) | 1 | ✓ | – | – | – | – | ✓ | – | – | – | – |
| MDCS Oh et al. (2022) | 2 | ✓ | – | ✓ | – | – | ✓ | – | – | – | – |
| **AIDE Yang et al. (2023a)** | 4 | ✓ | – | – | – | ✓ | ✓ | ✓ | ✓ | – | – |
| **Sage Deer** | 4 | ✓ | ✓ | ✓ | ✓ | ✓ | ✓ | ✓ | ✓ | ✓ | ✓ |

## 2 RELATED WORK

**Multi-modality Large Model in Driving**  Multi-modality large models are widely used in autonomous driving, such as automatic planning and control Mao et al. (2023); Cui et al. (2023a), perception Wang et al. (2020), and driver health monitoring Hecht et al. (2018) among others. By aggregating multi-modal information (e.g., vision, speech, point cloud, etc.) Yang et al. (2023b), multi-modal large models can more effectively help autonomous driving systems perceive the scene inside and outside the vehicle, and make more intelligent decisions.

In the vehicle's user-facing interface, the driver's emotions, health, posture, and actions can help the autonomous driving system interact dynamically and adapt to various driving styles. Cui et al. Cui et al. (2023b;a) first proposed to precisely enhance the decision-making process of autonomous driving by perceiving multi-modal information inside and outside the vehicle. Although these methods can perceive multi-modal information, they adopt a single strategy for different users and cannot make adaptive decisions for different users. Subsequent improvements Cui et al. (2023c); Yang et al. (2024) attempt to promote fine-grained understanding of user language and enhance the control performance of the vehicle system, but these methods overlook the impact of hidden emotions, health, and action information on vehicle safety and efficiency.

In this paper, we super-align various modal information of users, which not only includes the user's own interaction, but also perceives the user's health, emotions, posture, behavior, and other information, thereby enhancing the intelligence of the vehicle system.

**Retrieval-Augmented Generation**  Retrieval-Augmented Generation (RAG) enhances the accuracy of knowledge-intensive tasks by retrieving relevant information from an external knowledge base, which can be continuously updated Gao et al. (2023). Specifically, RAG first collects external information and saves old user queries Ma et al. (2023), organizes the knowledge, aligns it, and stores it in a repository provided by the user. Then, an efficient index is built to retrieve image data. Afterward, RAG retrieves internal information based on the prompts provided by the user, which mainly relies on priority ranking and retrieval. Finally, RAG merges the proposed query and selected information inputs it into a large model and returns a more accurate response. Based on this, RAG can help answer questions that vary according to specific needs while also aggregating the dialogue history scenario, enabling the model to remember customized information effectively.

In this paper, we use RAG to store customized information of users, including their driving habits and behaviors, which helps build a more human-centered vehicle driving agent, and can use customized requirements of specific users.

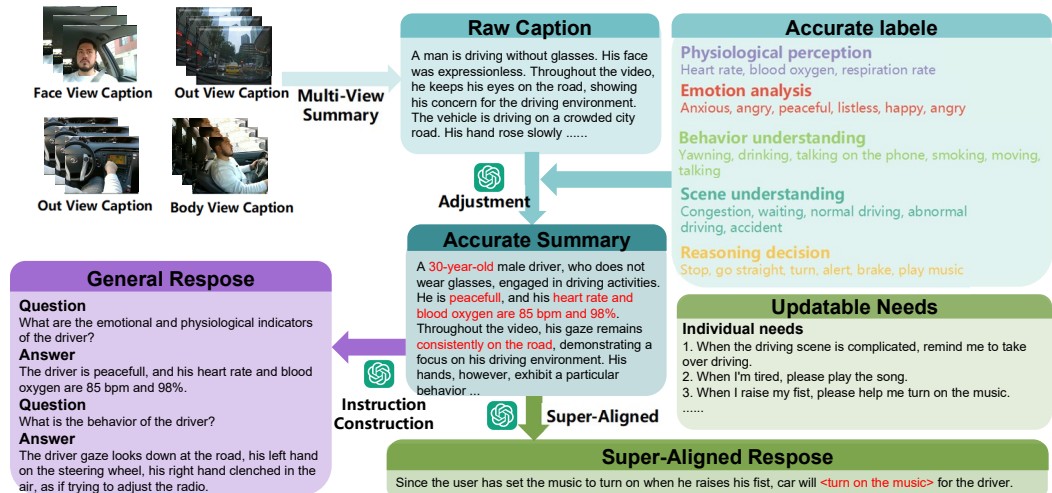

Figure 2: (a) We use existent multimodal model tools to generate captions for the videos likes (Li et al., 2023a; Zhang et al., 2023a). (b) Then, we set a reasonable prompt to merge the information from different videos. (c) We took advantage of the existing tags (including physiological indicators, emotional indicators, action indicators, behavioral indicators, scene understanding, and reasoning decision-making) to correct and supplement the captain. (d) Next, we use GPT4 (Achiam et al., 2023) as an assistant to build question-answering pairs for different tasks (including physiological indicators, emotion, behavior, and so on.). (e) We design multiple user preferences, and GPT4 responds to the current scenario based on user preferences.

## 3 DATA CURATING AND BEACHMARKING

### 3.1 DATA COLLECTION

Our goal is to create a driving generalist (physiological estimation, emotional estimation, gesture estimation, body motion estimation, driving behavior estimation, driving behavior detection, and driving decision-making) in the intelligent cockpit. We selected three of the most recent multi-view multi-task driving datasets AIDE and DMD. In addition, we use the latest contactless physiological measurement technology to monitor the user's health, which can estimate heart rate and blood oxygen only with a camera. We collected 5 datasets (VIPL-HR, V4V, PURE, BUAA-rPPG and UBFC).

### 3.2 GENERAL INSTRUCTION CONSTRUCTION

To construct natural language descriptions for sage deer, we use existent multimodal model tools (InternVideo (Wang et al., 2022), Tag2Text (Huang et al., 2023), or GRiT (Wu et al., 2022)) to generate captions for the frame sampled at equal intervals automatically. Likes (Li et al., 2023a; Zhang et al., 2023a), we set the reasonable prompt to merge the information of different frames. Existing image caption methods are often inaccurate or insufficiently annotated for an intelligent cockpit system. So, we took advantage of the existing tags (including physiological indicators, emotional indicators, action indicators, behavioral indicators, scene understanding, and reasoning decision-making) in the dataset to correct and supplement the captain. Next, we use GPT4 (Achiam et al., 2023) as an assistant to build question-answering pairs for different tasks. We used GPT4 to summarize, fuse, and correct these captains. See supplementary materials for details.

### 3.3 SUPER-ALIGNED REACTION

Users have different needs for driving cockpit, especially interactivity and trustworthiness. Interactivity means creating gestures, emotions, and body movements and tailoring cockpit feedback to user needs. Trustworthiness refers to personalized warning feedback on user fatigue, bad mood, and bad behavior. For this purpose, we use GPT4 to design the needs of different users and give specific feedback according to the needs of users and the information of the current scenario.

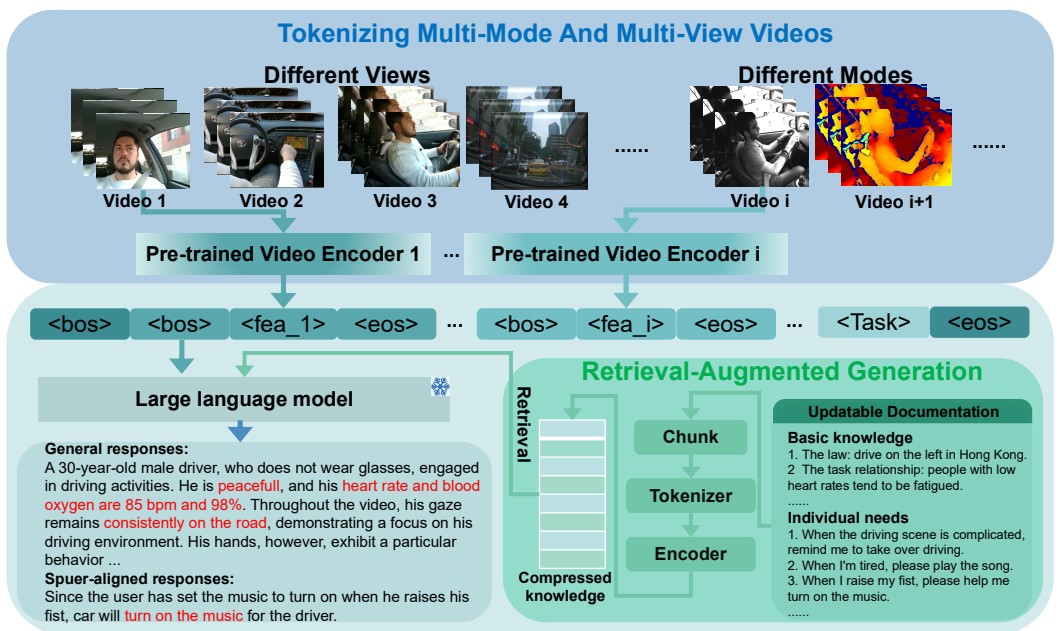

Figure 3: Overview of our framework. RGB, depth and NIR videos of different modes are extracted by a pre-trained encoder and converted into tokens. These tokens concatenate problems and retrieve enhanced linguistic features. Where the search augmentation generated to language features are now features that will be generated by the interchangeable document, via chunk, tokenizer, and further encoder. Ultimately, the LLM will respond based on the updatable document content and video content.

## 4 METHOD

We build a retrieval-augmented framework to build a super-aligned and generalist agent in the cockpit. This framework has three strengths: (1) Any mode and view video tokenization. (2) Generalists understand. (3) Super-aligned reactions.

### 4.1 TOKENIZING MULTI-MODE AND MULTI-VIEW VIDEOS

For driving cockpit, leveraging multiple modes and perspectives can significantly enhance the model's ability to interpret complex scenes, especially under challenging conditions such as poor weather or low-light environments. This subsection outlines our approach to tokenizing multi-mode and multi-view videos, ensuring that the model effectively integrates diverse visual information and physiological indicators.

**Tokenizing Multi-Model.** To robustly monitor user and scene information under adverse conditions, we incorporate multiple sensing modalities, including RGB, near-infrared (NIR), and depth cameras. We uniformly use the ImageNet pre-trained ResNet18 as each frame rate feature extractor, and then we cat all the frames together to get the video features. Finally, a simple two-layer linear layer is mapped to the language space $em_{face} \in C \times L$. $C$ is the number of channels characteristic of the language model, and $L$ is a hyperparameter representing how many tokens are used to present the video features. In particular, to better understand the physiological indicators of faces in the video. To allow LLM to distinguish between different modes, we added identifiers for visual features. For example, add corresponding start and end symbols to the front and back of a face RGB video embedding $em_{rgb} \in C \times L, < RGB\ bos > em_{rgb} < RGB\ cos >$. Other videos are processed the same way and then fed into the LLM.

**Tokenizing Multi-View.** Input of multiple perspective information is necessary, such as viewing information around the entire vehicle or monitoring the scene and driver at the same time. Similarly to multi-model, we uniformly use the ImageNet pre-trained ResNet18 as the feature extractor for each

frame rate. This is followed by mapping a simple two-layer linear layer to the language space. Then, we also use perspective markers to mark different perspectives, for example, $em_{front} \in C \times L$, $< Front\ RGB\ bos > em_{front} < Front\ RGB\ cos >$.

**Tokenizing Physiological Information.** Through a contactless video of a person's face, we can obtain physiological indicators such as heart rate, blood oxygen and respiration. But directly feeding a long video into our model is very low effect. Following (Lu et al., 2023), we first convert the video into a spatio-temporal representation (STMap) and then extract the features with ResNet18 and convert them into tokens $em_{phys} \in C \times L$ through two linear layers. Similarly, we've added identifiers, $< Phys\ bos > em_{phys} < Phys\ cos >$

**Final representation.** In addition, the user's questions are concated with vision tokens and sent to the LLM. The losses and training strategy of our visual instruction tuning are the same as those of MLLM (Li et al., 2023a; Zhang et al., 2023a; Muhammad Maaz & Khan, 2023).

## 4.2 RETRIEVAL-AUGMENTED GENERATION

The training of large language models is very expensive, and it is easy to lose the ability to generalize in a small training data. User needs for a smart cockpit are often very different, but we can't fine-tune the LLM to each user's specific preferences. To do this, we have adopted a retrieval-augmented generation framework ash shown in Fig. 3.

Specifically, we built an updatable document. This document includes some basics and personal requirements. Then we chunk the document, here we use a very concise way to divide the document by sentences. Each sentence is then tokenized, and then the sentence is the same length by filling in the tokens. A four-layer convolutional encoder is then used to compress the literal token into a 1/8-length feature. Then, we calculate the similarity between visual features and text features, and only retain the first N features with high similarity. Splicing into visual features. Then, we calculate the similarity between visual features and text features, and only retain the first N features with high similarity. These retrieved temporal features are spliced onto visual features.

## 4.3 EXPERT KNOWLEDGE FUSION

Reasonable knowledge and carefully crafted prompts can significantly enhance the potential of Large Language Models (LLMs), enabling them to generate more nuanced and context-aware outputs. In the context of the smart cockpit, there exists a strong and intricate relationship between various factors such as physiological indicators, emotions, behaviors, and the vehicle itself. Understanding these correlations is crucial for optimizing driver assistance systems, improving safety, and providing a more personalized driving experience. To this end, we conducted a systematic investigation of task correlations in collaboration with vehicle human factors experts, enabling us to explore these dependencies in greater detail.

The Relationship Between Physiological State and Emotion: Classical studies in psychophysiology (e.g., James-Lange, Schachter-Singer) have long established that physiological states, such as heart rate variability, skin conductance, and muscle tension, are closely linked to emotional responses. In the context of smart cockpits, real-time monitoring of physiological signals can provide valuable insights into the emotional state of the driver. For instance, elevated heart rate and increased skin conductance may be indicative of stress or anxiety, particularly in challenging driving scenarios. Recognizing this connection allows the system to adapt accordingly, offering calming feedback or assistance in real-time.

In addition, we investigated prior knowledge of other tasks, including: The Relationship Between Physiological State and Behavior,The Relationship Between Physiological State and Behavior, The Relationship Between Physiological State and Behavior. These relationships are not isolated but rather dynamically interact in real-world driving scenarios. By leveraging real-time data on cabin conditions — including environmental factors, driver behavior, and physiological/emotional states — the smart cockpit can generate a wider range of more accurate and contextually appropriate responses. This dynamic adaptation not only enhances the comfort and safety of the driver but also improves the overall driving experience. For example, in high-stress situations, the system may offer enhanced alerts or autonomous driving support, while in calm conditions, it might prioritize fuel efficiency or offer entertainment options.

Table 2: Baseline compared the performance of our approach on the super alignment protocol. Baseline is to splice user requirements directly into visual features and feed them into the LLM, taking advantage of the long-range modeling capabilities of the LLM itself.

| Method | AIDE | | DMD | |
|---|---|---|---|---|
| | BLEU | SPICE | BLEU | SPICE |
| **Baseline** | 0.184 | 0.301 | 0.191 | 0.296 |
| **Ours** | **0.214** | **0.315** | **0.204** | **0.305** |

By embracing the interconnected nature of these factors, intelligent systems in smart cockpits can better understand the human-driver interaction, anticipating needs and providing tailored responses that enhance both safety and the overall driving experience. Importantly, this approach underscores the potential of LLMs when incorporated with domain-specific knowledge, allowing them to generate more context-sensitive and accurate outputs in real time. Integrating such knowledge-driven, prompt-based systems into smart cockpits represents a significant leap in creating more intuitive, responsive, and adaptive vehicle environments.

## 5 EXPERIMENTS

### 5.1 DATASETS.

We found two recent data sets of very comprehensive annotations for assisted driving. The DMD is a driver monitoring dataset, an extensive dataset that includes real and simulated driving scenarios: distraction, eye distribution, drowsiness, handwheel interaction, and contextual data, in 41 hours of RGB, depth, and infrared video from 3 cameras, capturing the faces, bodies, and hands of 37 drivers. AIDE proposes an assisted driving awareness dataset that takes into account contextual information both inside and outside the vehicle in a natural scenario. AIDE enables overall driver monitoring through multi-perspective Settings of the driver and the scene, multi-modal annotations of the face, body, posture, and gestures, and four practical task designs for driver understanding. we collect five rPPG face video datasets ( VIPL-HR Niu et al. (2019), PURE Stricker et al. (2014), UBFC-rPPG Bobbia et al. (2019), V4V Revanur et al. (2021), and BUAA-MIHR Xi et al. (2020)), mostly with subjects remaining still, and some with head movements.

### 5.2 TRAINING DETAILS.

We tain our model on A6000 for 2 epochs. The learning rate, and weight decay are set to 0.001, and 0.02, respectively. The maximum sentence length is set to 64. That is, if the sentence is too long, excess parts will be discarded, and if the sentence is too short, 0 will be filled to make the length uniform. All image vision encoders use the pre-trained resnet 18 on imagenet as the pre-trained model, and these vision encoders are fine-tuned. For the feature extraction of physiological signals, we use the pre-trained model of NEST-rPPG Lu et al. (2023) and do not fine-tune it.

### 5.3 BASELINES AND EVALUATION METRICS

We compare our proposed framework's anomaly understanding performance with SOTA video understanding baselines. We select five baselines: Video-ChatGPT (Muhammad Maaz & Khan, 2023), VideoChat (Li et al., 2023a), Video-LLaMA (Zhang et al., 2023a), LLaMA-Adapter (Zhang et al., 2023b), and Video-LLaVA (Lin et al., 2023). Our comparison aims to determine whether these baselines can fully understand and interpret video anomalies.

To accurately evaluate our model's performance, we adopt BLEU Bilingual Evaluation Under-study(BLEU) and SPICE (Papineni et al., 2002) to measure word overlap between the model-generated text and the ground truth. This approach enables us to objectively assess the similarity and consider various levels of granularity at the text level, thus clearly indicating how well the model understands and describes anomalies.

Table 3: Generalist capabilities on AIDE data sets.

| Method | Emotion | | Behavior | | Scene | | Condition | |
|---|---|---|---|---|---|---|---|---|
| | BLEU | SPICE | BLEU | SPICE | BLEU | SPICE | BLEU | SPICE |
| **Video-ChatGPT** | 0.200 | 0.280 | 0.170 | 0.205 | 0.190 | 0.330 | 0.195 | 0.320 |
| **VideoChat** | 0.205 | 0.285 | 0.175 | 0.210 | 0.195 | 0.335 | 0.200 | 0.325 |
| **Video-Llama** | 0.217 | 0.312 | 0.183 | 0.223 | 0.207 | 0.352 | 0.211 | 0.337 |
| **Llama-Adapter** | 0.215 | 0.310 | 0.190 | 0.225 | 0.210 | 0.340 | 0.189 | 0.321 |
| **Video-LLaVA** | 0.202 | 0.290 | 0.172 | 0.205 | 0.190 | 0.330 | 0.195 | 0.320 |
| **Ours** | **0.232** | **0.331** | **0.194** | **0.242** | **0.225** | **0.369** | **0.223** | **0.360** |

Table 4: Generalist capabilities on DMD data sets.

| Method | Action | | Gaze | | Hand | |
|---|---|---|---|---|---|---|
| | BLEU | SPICE | BLEU | SPICE | BLEU | SPICE |
| **Video-ChatGPT** | 0.175 | 0.250 | 0.150 | 0.185 | 0.160 | 0.310 |
| **VideoChat** | 0.190 | 0.260 | 0.160 | 0.185 | 0.160 | 0.310 |
| **Video-Llama** | 0.205 | 0.275 | 0.170 | 0.205 | 0.180 | 0.325 |
| **Llama-Adapter** | 0.202 | 0.265 | 0.170 | 0.201 | 0.175 | 0.320 |
| **Video-LLaVA** | 0.195 | 0.295 | 0.155 | 0.220 | 0.170 | 0.315 |
| **Ours** | **0.235** | **0.315** | **0.195** | **0.230** | **0.210** | **0.340** |

## 5.4 GENERALIST PERFORMANCE

Our model can estimate the driver's emotion, physiological indicators, gaze, physical behavior, hand behavior, driving scene and vehicle state. In order to more clearly evaluate the ability of the model in different subtasks, we conducted systematic evaluation on two multi-task datasets, fatigue and physiological indicators. First, we used the default dataset partitioning on the AIDE dataset to train and test the results shown in Table 3. As you can see from the table, our algorithm has improved tremendously. This is mainly because, using expert knowledge, LLM can better integrate the connections between tasks.

Further, we verify our algorithm on DMD data set. As shown in Table 4, our algorithm achieves excellent performance on all tasks. All these results prove the effectiveness of our algorithm. Of course, our algorithm is very concise, only through RAG to retrieve the relevant expert knowledge to improve the accuracy of the reply. Further, we give an example of our model's dialogue to illustrate the generalist power of our model 4.

## 5.5 SUPER-ALIGNED PERFORMANCE

Different users have different needs for driving a smart cockpit, which requires us to super-align our users' biases. One of the simplest ways is to put all the user's preferences in the form of text at the top of the LLM input, and then use the LLM's own long-distance modeling capabilities to bias it. This most straightforward approach is considered our Baseline. Our approach is to complete the super alignment with a learned RAG. The performance of these two methods is shown in Table 2.

As shown in Table 2, our approach has improved dramatically. This shows that LLM itself has a limited ability to understand long texts, and it is difficult to make different responses to the results according to its visual features and prior knowledge. This explains the need for our RAG framework.

## 6 CONCLUSION

In this study, we successfully curated and benchmarked comprehensive data to develop Sage Deer, a super-aligned and generalist driving agent for intelligent cockpits. Our data collection strategy integrated three state-of-the-art multi-view and multi-task driving datasets—AIDE, DMD, VIPL-HR,

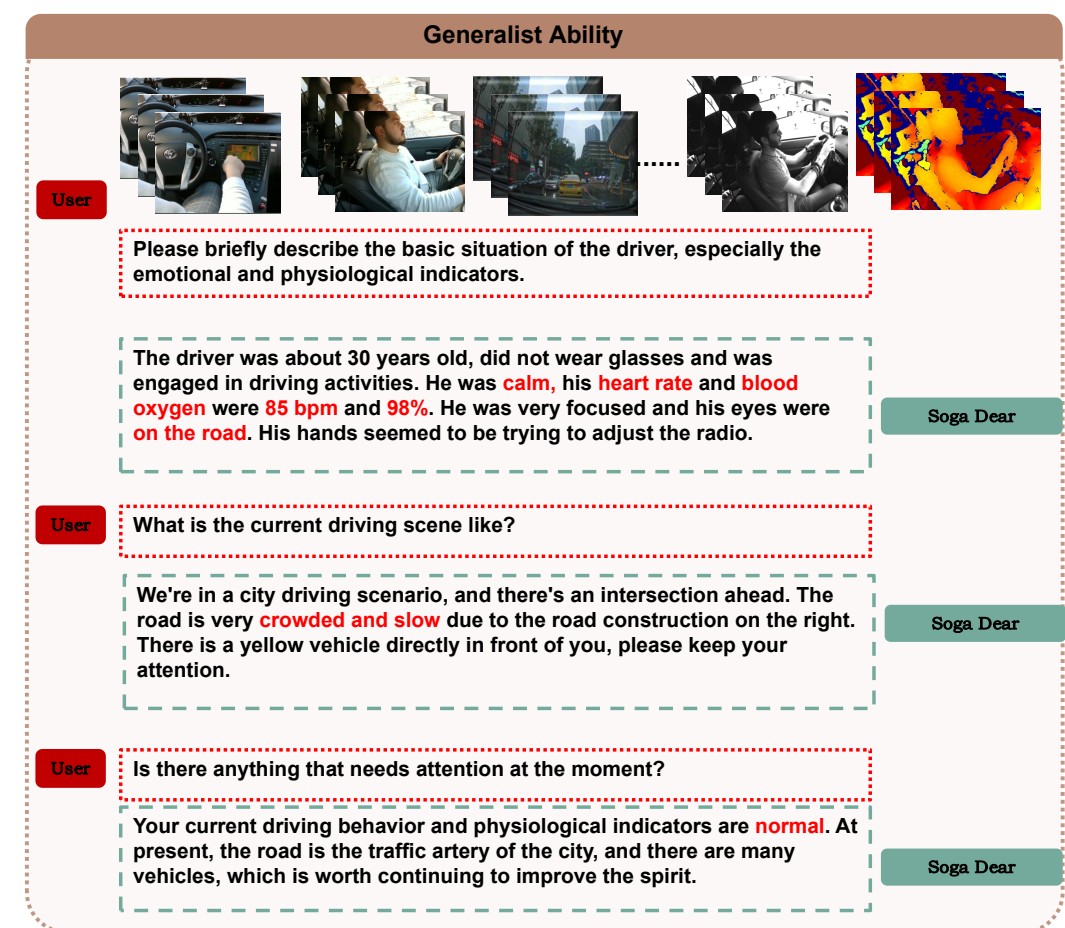

Figure 4: Generalist ability. Soga deer can reply to different tasks and has the ability to reply to an open domain.

V4V, PURE, BUAA-rPPG, and UBFC—alongside cutting-edge contactless physiological measurement technologies to monitor vital health indicators such as heart rate and blood oxygen levels using only camera inputs. To construct robust natural language instructions, we employed advanced multimodal models (InternVideo, Tag2Text, GRiT) for automatic video captioning, which were then refined and enhanced with existing tags and supplemented through GPT-4-generated question-answer pairs tailored to various driving-related tasks. Additionally, we addressed diverse user needs by leveraging GPT-4 to design personalized interactions, ensuring that the intelligent cockpit could provide tailored feedback based on individual preferences and real-time scenarios. Our benchmarking framework evaluates the agent's ability to interpret physiological data, emotional states, gestures, body movements, driving behaviors, and decision-making processes across multiple modalities. The integration of retrieval-augmented generation and expert knowledge fusion within our framework ensures that Sage Deer delivers accurate, context-aware, and personalized responses, significantly advancing the capabilities of intelligent driving cockpits in enhancing driver comfort, safety, and interaction.

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

## A  APPENDIX

You may include other additional sections here.

