# OpenReview forum: "A Super-Aligned Driving Generalist Is Your Cockpit"
_ICLR.cc/2025/Conference — ICLR 2025 Conference Withdrawn Submission_

### Official Review · Reviewer_ZJwV · 2024-10-30

**Soundness:** 3
**Presentation:** 3
**Contribution:** 2
**Rating:** 5
**Confidence:** 3

**Summary:**

The paper proposes to leverage multimodal data and LLMs to understand driver physiology, emotions, and behaviors in real-time. The authors use a RAG framework combined with expert knowledge integration to provide personalized feedback.

**Strengths:**

1、The paper introduce an interesting problem, emphasizing the importance of individual preferences in enhancing the driving experience.
2、The RAG framework addresses the need for flexible, personalized responses without extensive model fine-tuning.
3、The proposed method was evaluated on multiple driving datasets for its generalist and super-aligned performance.

**Weaknesses:**

1、The paper shows limited novelty, like RAG, are pre-existing approaches.
2、The "Expert Knowledge Fusion" section  isn’t clearly explained. Adding pseudocode or a flowchart could make it easier to follow.
3、The paper lacks ablation studies to verify the effectiveness of individual modules, such as physiological indicators and expert knowledge fusion.

**Questions:**

1、Can you explain why the paper chose ResNet18 as the pre-trained model instead of a more powerful option?
2、With multimodal data and personalized preferences present, how does RAG decide which information to prioritize for retrieval? Is there a specific prioritization or weighting system?
3、Could Sage Deer be compared more thoroughly with other recent intelligent driving agents, like DriveGPT or DriveLM, to provide a deeper understanding of its performance?

---

### Official Review · Reviewer_ZXDa · 2024-10-31

**Soundness:** 2
**Presentation:** 2
**Contribution:** 2
**Rating:** 6
**Confidence:** 3

**Summary:**

The paper introduces "Sage Deer," an intelligent driving cockpit system aimed at meeting personalized user needs through a multi-modal framework and a retrieval-augmented generation mechanism.

**Strengths:**

1. The concept of Sage Deer as a super-aligned, generalist agent offers a fresh approach to intelligent cockpit systems, adapting in real-time to individual user preferences.
2. The tailored application of a retrieval-augmented generation framework for the driving domain is a notable contribution, enabling efficient and adaptive responses to evolving user needs.
3. The development of a large-scale benchmark using a variety of datasets (AIDE, DMD, and others) to assess the system's decision-making and perception capabilities adds rigor and depth to the system’s evaluation.

**Weaknesses:**

1. How are the various inputs (e.g., visual, physiological) integrated to influence real-time driving decisions?
2. Could the paper delve deeper into how user interactions are managed, especially in complex scenarios? Are there any limitations to the system’s ability to interpret nuanced or less common user behaviors?
3. There are a few errors: for example, the purpose of "s" in lines 50 and 53 is unclear, “Out View Caption” is duplicated in Figure 2, and “Accurate labele” contains a spelling error.

**Questions:**

see weakness

---

### Official Review · Reviewer_tfDg · 2024-10-31

**Soundness:** 1
**Presentation:** 1
**Contribution:** 2
**Rating:** 1
**Confidence:** 4

**Summary:**

This paper presents a multi-modal LLM designed for human and scene understanding in autonomous driving. It integrates multi-view image and multi-modal inputs, using a retrieval-augmented generation strategy to improve test-time adaptation. For evaluation, a multi-modal, multi-view dataset for driving perception is introduced. The proposed method outperforms standard multi-modal LLM models.

**Strengths:**

1. The motivation to combine 3D scene perception with both visual and physiological signals from humans is clear and compelling. However, an ablation study on each signal and modality would enhance understanding of their individual contributions.
2. A dataset is established, incorporating multi-modal, multi-view sensor data along with QA pairs for evaluating the LLM’s scene understanding and reasoning capabilities.
3. The proposed method shows improved overall performance on the provided dataset.

**Weaknesses:**

Overall, I regret to say that this submission appears of low quality, with numerous errors suggesting it was submitted without thorough proofreading—a potential disservice to the reviewers.

1. There are numerous typographical and formatting errors, including typos, incorrect notations, capitalization issues, and incomplete sentences. Examples include:
   - L050: "technologies, s possess"
   - L053: '"with s,"'
   - L208: "supplement the captain"
   - L261, L265: "the language spaceemface ∈ C × L.", "emrgb ∈ C × L, < RGB bos > emrgb < RGB cos >."
   - L271, L277: "emfront ∈ C × L,"
   - L288: "framework ash shown in Fig. 3"
   - L307: "The Relationship Between Physiological State and Emotion: Classical"
   - L315-L317: "other tasks, including: The Relationship Between Physiological State and Behavior…" (repeated thrice)

2. The proposed method lacks novelty, as it is essentially a multimodal LLM with RAG, without any specific design tailored for the target task. Additionally, key methodological details, such as training strategies, specific model architectures, and hyperparameters, are missing.

3. Experimental analysis is limited. In-depth experimentation and analysis are needed to substantiate the claimed benefits of using a multimodal approach.

4. The dataset setup is unclear. Since the captions are generated by open-source VLMs, please clarify the measures taken to ensure their quality.

5. The related work citations do not consistently support the claims made. For instance, L308 references "Classical studies in psychophysiology (e.g., James-Lange, Schachter-Singer)…” without sufficient context.

6. The appendix section is empty. Please remove the placeholder text: "You may include other additional sections here."

7. Finally, as the dataset includes human subjects, please provide an ethics statement to address concerns regarding its use.

**Questions:**

Please see weaknesses section.

Additionally, real-world dataset construction rarely captures abnormal behaviors. How, then, does training on the proposed dataset support effective human behavior anomaly detection?

**Details Of Ethics Concerns:**

I recommend this paper be flagged for an ethics review due to concerns related to the proposed dataset, which includes human subjects. Key ethical considerations include:

1) The dataset's inclusion of human data raises concerns about compliance with copyright laws, data protection standards, and consent protocols under regulations like GDPR.

2) The use of human data requires careful consideration of ethical research practices, including whether informed consent was obtained, how the data will be stored, and the responsible handling and potential release of this data.

To ensure an ethically sound review, an ethics reviewer with expertise in privacy, legal compliance, and responsible research practices would be most suitable.

---

### Official Review · Reviewer_E6VX · 2024-11-04

**Soundness:** 3
**Presentation:** 2
**Contribution:** 2
**Rating:** 3
**Confidence:** 4

**Summary:**

This research presents "Sage Deer," an innovative super-aligned and generalist driving agent designed to enhance intelligent cockpit systems. The proposed framework addresses the challenges of personalized driving experiences and comprehensive omni-modal information processing through several key innovations.

**Strengths:**

Strength:

-	Seamlessly integrates data from various sensors (RGB, NIR, depth cameras) and multiple perspectives, enabling comprehensive environmental understanding.

-	Employs a unique mechanism that combines an updatable knowledge base with a large language model, enabling contextually relevant responses without extensive fine-tuning.

**Weaknesses:**

Weakness:
-	The authors should provide more comprehensive information about the model architecture, including specifics such as the choice of LLM with its size and so on.

-	In Figure 3, a "Pre-trained Video Encoder" is depicted, whereas Section 4.2 mentions the use of an "ImageNet pre-trained ResNet18." Are these referring to the same component? Additionally, how does this encoder handle other modalities? Lastly, how many tokens does the encoder output? Providing more detailed explanations would enhance understanding.

-	In Section 4.1, the author introduces specific start and end symbols to denote different modalities. Are these symbols newly added special tokens to the LLM's vocabulary? If so, how are these tokens initialized? Since the LLM remains frozen and is not further trained, how does the pretrained model recognize these new tokens?

-	In Section 5.2, the maximum sentence length is set to 64. How was this value determined? Since text sentences are processed by a tokenizer, why not base this parameter on the number of tokens instead? Were any experiments conducted to evaluate the impact of this choice on performance or the training and inference computational budget?

-	The sequence of tables and figures should be adjusted for consistency. For instance, Table 2 is only mentioned in Section 5.5, while Tables 3 and 4 are referenced earlier in the document before Table 2.

-	The manuscript requires improved writing quality, as numerous typographical errors are present. For example, on line 414, "model" should be corrected to "figure," and on line 261, a space is needed between the text and the equation.


The manuscript currently contains several typographical and writing errors, as well as some missing details, which is not ready for submission. I believe it would benefit from further revisions to address these issues and ensure it meets the standards required for submission to ICLR.

**Questions:**

see weakness

---

### Official Review · Reviewer_Ara9 · 2024-11-04

**Soundness:** 2
**Presentation:** 1
**Contribution:** 3
**Rating:** 3
**Confidence:** 3

**Summary:**

This paper aims to build a super-aligned and generalist driving agent, called sage deer for the intelligent driving cockpit. A new dataset is constructed for many tasks, e.g., physiological estimation, emotional estimation, gesture estimation, body motion estimation, driving behavior estimation, driving behavior detection, and driving decision-making. An MLLM is trained for unified tasks.

**Strengths:**

- The paper is the first to construct a unified dataset for MLLMs in intelligent driving cockpit. A multi-task dataset is provided and an MLLM is trained on the dataset.

**Weaknesses:**

- The dataset construction part is extremely lacking in details, including data curation, GPT4 labeling, etc. The paper states in many places that the details are in supplementary materials, but the supplementary materials have not been submitted. Besides, the contribution of "An intelligent driving cockpit super alignment evaluation protocol involving generalization ability for different needs was established" cannot be well-established in the paper.
- The qualitative results are very limited. Only in Fig. 4, some conversations are provided, and from this figure, we cannot know the full ability of the model.
- The writing of the paper was very hasty. many sentences are not clear and typos are everywhere, e.g., "serves as the interface for human interaction with s" in L053, and "Tokenizing Multi-Model" in L257.

**Questions:**

See weaknesses. Please provide more details as much as possible.

---

### Official Review · Reviewer_s9po · 2024-11-04

**Soundness:** 2
**Presentation:** 2
**Contribution:** 3
**Rating:** 5
**Confidence:** 5

**Summary:**

The paper introduces Sage Deer, a multi-modal, multi-view framework for intelligent driving cockpits, designed to provide personalized, context-aware assistance. It integrates RGB, NIR, and depth cameras, and captures diverse data on driver states such as physiology, emotion, and behaviour which enables comprehensive monitoring and real-time response. This data is processed through a language model, allowing for nuanced comprehension and interaction capabilities.

The system’s architecture relies on three core components: retrieval-augmented generation (RAG), multi-modal fusion, and expert knowledge incorporation. RAG allows Sage Deer to retrieve relevant external information, tailoring responses to user preferences. Multi-modal fusion combines data from various camera views, enhancing the model's understanding of the environment and driver states. Expert knowledge fusion further refines Sage Deer’s outputs by integrating specialized insights into physiological and emotional monitoring, optimizing its response relevance and accuracy.

Experimental results demonstrate Sage Deer’s effectiveness in multitasking and adapting to diverse user needs, providing a benchmark for intelligent cockpit design. By aligning AI capabilities with user-centered safety requirements, Sage Deer advances the potential of personalized driver assistance systems, positioning itself as a foundational technology for future ADAS applications.

**Strengths:**

Sage Deer integrates multi-modal and multi-view data sources, combining RGB, NIR, and depth cameras to achieve a highly adaptive and personalized intelligent cockpit system. The model’s use of Retrieval-Augmented Generation (RAG) allows it to pull relevant context-specific information from external sources, enhancing the system’s real-time responsiveness and ability to deliver highly accurate, personalized interactions aligned with individual driver preferences. This capacity for personalization goes beyond standard cockpit systems, as Sage Deer monitors physiological, emotional, and behavioural states to tailor responses to the driver's unique profile, significantly boosting both user engagement and safety.

The fusion of diverse sensor data enables Sage Deer to accurately perceive and interpret complex, dynamic conditions within and outside the vehicle, making it capable of maintaining performance under varying lighting and environmental scenarios. Its robust, real-time capabilities show substantial potential for practical applications in ADAS, offering intelligent, responsive support that adapts continuously to real-world challenges. Sage Deer’s architecture sets a new standard for intelligent cockpit systems, bringing together advanced AI components to enhance driver experience and overall vehicle safety in ways that align with the evolving demands of autonomous and semi-autonomous vehicles.

**Weaknesses:**

The paper needs improvement in writing. There are mistakes and citation errors. See at the end of this message.

The main issue is the novelty. It seems to combine multiple models to improve intelligent driving. This contribution is not good enough for a conference like ICLR.

In (Section 4.3), the authors mentioned that the model relies on visual tokenization of physiological data, such as heart rate and blood oxygen levels, to infer emotional or behavioural states. However, this approach assumes direct correlations with emotions, potentially leading to inaccuracies. While the authors have cited studies in psychophysiology suggesting links between physiological signals and emotions, the real-world application requires greater nuance. Authors should discuss impact on signals due to factors like individual baselines, environmental conditions, and physical activity.

The model’s use of a pre-trained ResNet18 for tokenizing RGB, NIR, and depth inputs may lack the capacity to capture the complex nuances needed for an intelligent cockpit system. To address this, the authors should conduct ablation studies comparing ResNet18 with advanced models like ResNet50, EfficientNet, ViT, and Swin Transformer to assess improvements in accuracy and robustness. Additionally, the current concatenation-based fusion strategy may underutilize the complementary data from multi-modal inputs. Testing different fusion techniques, such as attention-based and cross-attention methods, could identify more effective integration approaches. Further analysis of each modality’s impact would clarify the significance of RGB, NIR, and depth data, while transformer-based models could improve temporal understanding for tasks like fatigue tracking.

The reliance on a language model for contextual understanding may oversimplify dynamic driving scenarios, missing essential non-verbal cues for real-time safety. Ablation studies could address this by comparing language-only input to multi-modal input (e.g., visual, physiological, behavioural data) to assess non-verbal contributions to accuracy in safety-critical tasks. Testing each modality individually would highlight their impact while comparing the language model with and without RAG would clarify RAG’s role in context accuracy.

Writing:
Line 37: He -> it
Line 50: s possess?
Line 53: with s?
Line 61: reference a?
Line 66: repeat of Sima et al.
Line 188: Beachmarking -> Benchmarking

**Questions:**

1. The novelty of the contribution.
2. In Section 4.3, the model uses visual tokenization of physiological signals to infer emotions. How does the model account for individual differences in physiological baselines or external factors (e.g., physical activity, environmental conditions) that might affect these signals?
3. Why was ResNet18 chosen over more advanced models like ResNet50, EfficientNet, or transformer-based architectures? Did you conduct any initial tests with these models?
4. Would you consider performing ablation studies comparing ResNet18 with more powerful feature extractors to evaluate improvements in capturing behavioral and environmental nuances?
5. How does ResNet18 perform in capturing temporal dependencies in sequential data, particularly for tasks requiring context awareness over time, such as fatigue tracking?
6. Given the current use of a concatenation-based fusion approach, have you explored other fusion techniques, such as attention-based fusion or cross-attention mechanisms, to maximize the complementary data from RGB, NIR, and depth inputs? Have you considered ablation studies to evaluate the impact of each modality independently?
7. How does the model handle or prioritize input from non-verbal cues compared to language-based cues in dynamic driving contexts?

---

### Note · Authors · 2024-11-15

**Comment:**

Thank you for your effective suggestions. We will further improve the quality of our papers with these suggestions.

**Withdrawal Confirmation:**

I have read and agree with the venue's withdrawal policy on behalf of myself and my co-authors.